# Evaluation of Atypical Chemokine Receptor Expression in T Cell Subsets

**DOI:** 10.3390/cells11244099

**Published:** 2022-12-16

**Authors:** Messias Oliveira Pacheco, Fernanda Agostini Rocha, Thiago Pinheiro Arrais Aloia, Luciana Cavalheiro Marti

**Affiliations:** Hospital Israelita Albert Einstein, Rua Comendador Elias Jafet, São Paulo 05652-000, Brazil

**Keywords:** chemokines, atypical chemokine receptors, ACKR, T cell, memory, naïve

## Abstract

Chemokines are molecules that pertain to a family of small cytokines and can generate cell chemotaxis through the interaction with their receptors. Chemokines can trigger signaling via conventional G-protein-coupled receptors or through atypical chemokine receptors. Currently, four atypical chemokine receptors have been are described (ACKR1, ACKR2, ACKR3 and ACKR4). ACKRs are expressed in various cells and tissues, including T lymphocytes. These receptors’ main function is related to the internalization and degradation of chemokines, as well as to the inflammation control. However, the expression of these receptors in human T lymphocytes is unclear in the literature. The objective of this study was to evaluate the expression of ACKRs in different subpopulations of T lymphocytes. For this, peripheral blood from healthy donors was used to analyze the expression of ACKR2, ACKR3 and ACKR4 by immunophenotyping CD4, CD8 T lymphocytes and, in their subsets, naive, transition and memory. Results obtained in this study demonstrated that ACKR2, ACKR3 and ACKR4 receptors were expressed by T lymphocytes subsets in different proportions. These receptors are highly expressed in the cytoplasmic milieu of all subsets of T lymphocytes, therefore suggesting that their expression in plasma membrane is regulated after transcription, and it must be dependent on a stimulus, which was not identified in our study. Thus, regarding ACKRs function as scavenger receptors, at least for the ACKR3, this function does not impair the chemotaxis exert for their ligand compared to the typical counterpart receptor.

## 1. Introduction

Chemokines belong to a large family of small cytokines engaged into the process of immune cells migration and residence [1,2]. Chemokine receptors are known as a superfamily of chemokine G-protein coupled receptors (GPCRs). They control the immune cells behavior to promote chemotaxis, cell adhesion, and mediators release [1,2].

The signaling is followed by the interaction of a chemokine and their receptor, which usually activates the G protein subunits [1,2]. Atypical chemokine receptors (ACKRs) are also cell surface receptors homologous to GPCRs, and upon ligation to their cognate ligand, they fail in inducing the classical signaling through G protein or promoting the downstream cellular responses. Instead, the signaling is described to be β-arrestin pathway-dependent, linked to internalization of receptor and their ligands, followed by the transport of chemokines to intracellular degradative compartments [3].

ACKRs are described to control chemokines bioavailability in the milieu and to have emerged as important contributors for the homeostasis maintenance during an inflammatory immune response [3,4,5].

Four types of atypical chemokine receptors have been described, ACKR1 to ACKR4 [1,3,6,7]. Two additional candidates designated as ACKR5/CCRL2 and ACKR6/PITPMN3 have also been identified, but they are awaiting further functional characterization [5]. To date, the literature mentions the expression of three ACKRs in leukocytes: ACKR2 [8] ACKR3 [7], and ACKR4 [9].

Briefly, ACKR2, also known as D6 or chemokine receptor-binding protein 2 (CCBP2), was identified in several lineage and mouse cells as endothelial cells, keratinocytes, placental trophoblasts, B and T cells [6,10,11,12,13,14]. ACKR2 ligands are chemokines CCL2-5 CCL7-8, CCL11-13, CCL17 and CCL22. The main function described for ACKR2 is to internalize their ligands and degrade them [5,8,15,16].

ACKR3, also known as chemokine receptor type 7 (CXCR7) or RDC1, was identified in several cells such as lymphatic endothelial cells, placental trophoblasts, and some leukocytes [5,7,17]. ACKR3 ligands are CXCL11 and CXCL12, the CXCL11 being a chemoattractant of T lymphocytes and an interferon inducer [9,18], while CXCL12, also known as stromal-derived factor 1 (SDF-1), is a chemoattractant for leukocytes and hematopoietic stem cells. The main function described for this receptor is to internalize their ligands and degrade them. ACKR3 was also reported signals through the interaction with their counterpart typical receptor CXCR4 [9,18,19,20].

ACKR4 is also known as CCRL1 and CCX-CKR [3,21,22]. This atypical receptor is expressed in various tissues such as heart, brain, spleen, intestine, endothelial cells, DCs, B and T lymphocytes [21,22,23]. ACKR4 ligands are CCL19, CCL21, CCL25 and CXCL13, and the ACKR4 plays a role in internalizing its ligands, and degrades them. ACKR4 was also described to interfere in the fate of B and T cells as well [23,24,25,26].

Regarding the protein expression of ACKRs in the membrane of human cells, the ACKR2 expression was only demonstrated in human lineage cells, and its expression was not described previously in human T cells. ACKR3 was found to be expressed on human T cells [17]; however, to date, there are no reports about its expression on human T cell subsets. In addition, the ACKR4 was only described in mouse T cells and human lineage cells, but this receptor was not previously described to be expressed in human T cells.

Given the importance of these receptors in controlling inflammatory immune responses, herein we have followed their expression on leucocytes by focusing in human T cells and their subsets. These three receptors are present in human T cells and they display variable expression on these cells’ surface. In addition, ACKRs are highly expressed in the human T cells’ cytoplasm, which suggests that their expression on these cells’ surface is controlled after protein translation. In addition, regarding the function of these receptors as scavenger receptors, at least for ACKR3, this study has confirmed that this function is not preventing the chemotactic function exert for their ligand CXCL12 compared to the typical counterpart receptor.

## 2. Materials and Methods

### 2.1. Samples Obtainment

This project was approved by the Ethical Committee of Hospital Israelita Albert Einstein under CAAE number 93808318.6.0000.0071. After all volunteers have signed an informed consent form, peripheral blood samples were obtained from 40 healthy individuals. These 40 samples were distributed among different experiments. The initial 8 samples were used in order to establish the antibody titration, compensation and to set up cell sorting experiments. Subsequently, 7 samples were freshly stained for the ACKRs, 10 samples were stained for ACKRs on T cells subsets CD4 and CD8, 7 samples were used for ACKRs on T cells subsets (memory/naive), 4 samples were used for surface and intracellular expression of ACKRs, 4 samples were used to test different stimuli, and finally, another 4 were used for cell sorting and chemotaxis.

### 2.2. Peripheral Blood Mononuclear Cell Preparation

Mononuclear cells from peripheral blood of healthy donors were separated by density gradient. Briefly, peripheral blood was diluted (1:1) in phosphate-buffered saline (PBS), and this mixture was transferred to a 15 mL conical tube containing 5 mL of Ficoll/Hypaque (GE Healthcare, Chicago, IL, USA). The tubes were centrifuged for 30 min at 500× *g* and room temperature without deceleration force (Centrifuge 5810/5810R, Eppendorf, Hamburg, Germany). The cell interface was removed and transferred to a new 15 mL conical tube, diluted in PBS, and centrifuged again for 5 min at 500× *g* and room temperature. After centrifugation, the supernatant was discarded and the mononuclear cells were resuspended with AIM-V (Gibco^®^, Carlsbad, CA, USA) supplemented with 10% human AB serum (Valley Biomedical, Winchester, VA, USA) and 200 mM L-Glutamine, Non-Essential Amino Acids, Hepes, Pyruvate, and 1% of Antibiotic-Antimycotic (Gibco^®^, Carlsbad, CA, USA). Next, these cells were selected by cell sorting or using magnetic beads.

### 2.3. Flow Cytometry Assays

After peripheral blood obtainment, 400 µL of whole blood was transferred to a cytometry tube, resuspended in 2000 µL of Excellyse Live (1×) lysing solution (Exbio, Praha, Czech Republic), and incubated at room temperature for 20 min for red blood cell lysis.

After lysis, the sample was centrifuged at 500× *g* for 5 min. The supernatant was discarded and the pellet resuspended in 500 µL of cytometry buffer (PBS supplemented with 0.1% human albumin and 0.1% sodium azide). For separated mononuclear cells or sorted cells, this lyses first step was unnecessary.

Then, cells were stained for surface with the following specific antibodies: CD45 (Alexa 700, clone HI30, BD Pharmingen, San Diego, CA, USA), CD3 (APC-H7, clone SK7, BD Pharmingen, San Diego, CA, USA), CD4 (BV-421, clone L120, BD Horizon), CD8 (PE-CF594, clone RPA-T8, BD Pharmingen, San Diego, CA, USA), D6/ACKR2 (FITC, clone 196124, R&D Systems, Minneapolis, MN, USA), CXCR7/ACKR3 (BV421, clone 10D1, BD Horizon), CCRL1/ACKR4 (BV605, clone 13E11, BD Biosciences, San Jose, CA, USA), and CD45RO (PE, clone UCHL1, BD Pharmingen, San Diego, CA, USA).

For intracellular staining, cells were stained for leucocytes markers described above, followed by fixation with Lysing Solution (BD Biosciences, San Jose, CA, USA) and permeabilization with Permeabilizing Solution 2 (BD Biosciences, San Jose, CA, USA), then labeled with anti-ACKR2, anti-ACKR3 and anti-ACKR4.

After the addition of the antibodies, the material was incubated for 20 min in a dark chamber. As controls, unlabeled cells were used, fluorescence minus one (FMO) and isotype control mouse BALB/c IgG2a, κ-FITC from BD Pharmingen, Mouse IgG1, κ-BV421 (clone:MOPC-21) and Mouse IgG2a, κ-BV510 (clone:MOPC-173) from Biolegend (San Diego, CA, USA).

The cells were acquired using the flow cytometer Fortessa–LSRII (BD Biosciences, San Jose, CA, USA). Data analyses were performed using FlowJo software version 10.6.2 (BD Biosciences, San Jose, CA, USA). MFI values were obtained from FlowJo Software for each AKCR fluorescence, and the background MFI from isotype control was diminished from the original value.

### 2.4. T Cell Selection Using Magnetic Beads

CD3 T lymphocytes were captured by positive selection using MicroBeads human CD3 kit (Miltenyi Biotec, Bergisch Gladbach, Germany). Briefly, mononuclear cells were washed in ice-cold column buffer (PBS with 0.5% human albumin and 2 mM EDTA) and labeled with a monoclonal antibody coupled to magnetic microbeads, with specificity for the ϵ chain of human CD3. The marked cells were selected, and cell purity was verified by flow cytometry using a fluorescent antibody CD3-PE-CF594 (clone OKT3 and UCHT1 BD Horizon). The purity of this selection was above 99%.

### 2.5. Cell Sorting

Peripheral blood T cells from healthy donors were stained with anti-CD3 (APC-H7, clone SK7, BD Pharmingen, San Diego, CA, USA), anti-ACKR3/CXCR7 (BV421, clone 10D1, BD Horizon) and anti-CXCR4/CD184 (PE, Clone 12G5, BD Pharmingen, San Diego, CA, USA) for cell sorting of 3 populations of T lymphocytes: ACKR3/CXCR7, CXCR4 and ACKR3/CXCR7:CXCR4. Sorting was performed at a temperature of 10–12 °C on the Moflo Astrio EQ cytometer (Beckman Coulter, Indianapolis, IN, USA) using a 100 µM nozzle tip, with a pressure of 25 psi. Purity mode was defined as sorting decision. Throughout the process, the separation efficiency was between 90 and 95% and the purity for 2 populations was over 90% (CXCR4 and CXCR4:CXCR7/ACKR3), and for ACKR3/CXCR7 the purity was 77.5%.

### 2.6. T Cell Activation by Stimuli

Previously selected T cells were resuspended in supplemented AIM-V, and placed in culture at a concentration of 5 × 10^5^ cells/mL per well into a 24-well culture plates. These cells were maintained in cultured in absence (control) or presence of 10 ng/mL of different stimuli: as CXCL8; IL-15, CCL20; IL-32 (all from R&D Systems, Minneapolis, MN, USA). These stimuli were chosen based on a previous data (not published) of mRNA expression on T cells treated with them. In addition, as positive control for activation, cells were also stimulated (1:1) with dynabeads anti-CD3:CD28 (Gibco—ThermoFisher, Waltham, MA, USA). The T cells were kept at 37 °C for 18 h, and further analyzed by flow cytometry for the protein expression of ACKR2, ACKR3 and ACKR4.

### 2.7. Transwell Migration Assay

After separations by cell sorting, T cells expressing CXCR4, or CXCR7/ACKR3 or both CXCR4:CXCR7/ACKR3 were quantified and cultured for cell migration assay using transwell inserts with 3 µM pores along a CXCL12 concentration gradient. To perform the assay, 1 × 10^5^ cells were resuspended in 300 μL of AIM-V culture medium and added to the upper wells of transwell inserts (Polyester Membrane, Corning Incorporated, Laredo, TX, USA), and 1000 μL of AIM-V culture medium containing 10 ng of CXCL12 were added to the bottom of the insert. The cultures were kept in a 5% CO_2_ incubator (for 24 h at 37 °C). After this period of culture, the cells that migrated to the lower chamber were removed, relabeled with antibodies for CD3, CXCR4 and CXCR7/ACKR3, and again evaluated by flow cytometry.

### 2.8. Confocal Imaging

Mononuclear cells from peripheral blood of healthy donors were separated by density gradient as previously described. Next, cytospin technique was performed, 50 µL containing 2 × 10^6^ cells in suspension were used to prepare the slides, followed by a centrifugation of 1000 rpm for 5 min. Cells were fixed with parafolmadehyde 4% for 30 min and permeabilized with triton X-100 (0.05%) before the staining process. Next, the slides were stained with CD3 (PE-CF594, clone:OKT3, BD Pharmingen) (1:20 dilution), D6/ACKR2 (FITC, clone 196124, R&D Systems) (1:50 dilution), and incubated overnight at room temperature. The slides were prepared for analysis with Fluormount-G containing DAPI for nuclei staining (e-Biosciences, San Diego, CA, USA) and covered with coverslips. These slides were analyzed by a confocal microscopy Zeiss 710 (Oberkochen, Germany).

### 2.9. ACKRs mRNA Evaluation on Database

We have consulted datasets for mRNA expression used to classify protein-coding genes into expression clusters for immune cells. The transcriptomics data was evaluated by assessing the site The Human Protein Atlas (https://www.proteinatlas.org) on 2 November 2022. The ACKR2, ACKR3 and ACKR4 mRNA expression on T lymphocytes populations was evaluated according to Schmiedel [27] and Monaco [28] datasets. The protein localization in subcellular compartments was also consulted in the Human Protein Atlas plataform.

### 2.10. Statistical Analysis

For statistical analysis, the GraphPad Prism program version 7.02 and 9.0 (GraphPad Prism Software Incorporation, San Diego, CA, USA) was used. When comparing two groups, the *t* student test was used and ANOVA with Bonferroni correction was used for multiple comparisons. The difference between data was considered significant when *p* < 0.05 was observed.

## 3. Results

### 3.1. Expression of ACKR2, ACKR3 and ACKR4 in Different Leukocyte Populations

Initially, the expression of atypical chemokine receptors ACKR2, ACKR3 and ACKR4 was evaluated in populations of leukocytes (CD45) from peripheral blood of healthy donors by flow cytometry. T cells were selected based on CD3 expression, while for the other leukocytes (monocytes, granulocytes and NK+B cells), the selection was based on their granularity and absence of CD3 expression (Figure 1A). The ACKR2, ACKR3 and ACKR4 evaluation was based on a control for every receptor in each population as displayed in Appendix A.

The results confirmed the expression of all three receptors on circulating leukocytes with variable distribution between individuals (Figure 1B). The monocyte population exhibited the higher expression of ACKR2, ACKR3 and ACKR4 compared with T and B cells/NK cells (Figure 1B) and corroborates median intensity of fluorescence (MFI) distribution among populations (Figure 1C). These data demonstrate that circulating T lymphocytes from healthy individuals express atypical chemokine receptors, ACKR2, ACKR3 and ACKR4, which comprises the targets of this study.

### 3.2. Subsets of T Lymphocytes That Express ACKR2, ACKR3 and ACKR4

The T lymphocytes were identified by the expression of CD45 and CD3, as well as their subsets by CD4 and CD8 (Figure 2A). The frequency of CD45/CD3 T lymphocytes expressing ACKR2 or ACKR3 was higher than those expressing ACKR4 (Figure 2B). This same profile was observed in the T lymphocyte subsets CD4 (Figure 2C) and CD8 (Figure 2D).

When the expression of atypical receptors was compared between the subsets of CD4 and CD8 T lymphocytes, the frequency of CD4 expressing ACKR2, ACKR3 or ACKR4 was significantly higher than the frequency CD8 expressing the same atypical receptors on their surfaces as shown in Figure 2E–G.

Taken together, these data demonstrated that despite the variation between individuals, both CD4+ and CD8+ T lymphocytes expressed ACKR2, ACKR3 or ACKR4, and CD4 T lymphocytes stood out in the frequency of cells expressing these receptors compared with CD8, regardless of the atypical receptor expressed on their surfaces.

### 3.3. Naive and Memory T Lymphocytes Express Atypical Chemokine Receptors

The expressions of ACKR2, ACKR3 and ACKR4 were evaluated in T lymphocyte subsets of memory (CD45RO+), naive (CD45RO-), and transition (CD45ROint) as demonstrated in Figure 3A.

For ACKR2, naive T lymphocytes expresses more of this receptor than transition and memory subsets (Figure 3B). CD4 T lymphocytes expressing ACKR2 were higher in the naive and memory subsets compared with transition subset (Figure 3C). The same was observed for naive CD8 T lymphocytes compared to transition or memory subsets (Figure 3D).

Among the T lymphocytes that express ACKR3, the naive ones stand out in frequency compared to transition and memory subsets, and the frequency of memory was higher than in the transition subset (Figure 3E). This difference was similar, but to a lesser extent in the CD4 ACKR3 T lymphocytes (Figure 3F). However, CD8 T lymphocytes that express ACKR3 were found at a lower frequency in the memory population than naive and transitional ones (Figure 3G).

As for the frequency of T lymphocytes that express ACKR4, the naive population also outperformed the memory and transition subsets, which were similar (Figure 3H). However, in CD4 T lymphocytes, the frequency of transition subsets expressing ACKR4 was lower compared with naive and memory subsets that also express this receptor (Figure 3I). In contrast, CD8 ACKR4+ T lymphocytes showed no difference in expression between naive, transition or memory populations (Figure 3J).

In general, naive T cells express ACKR2, ACKR3 or ACKR4 more than memory or transition ones. However, this characteristic is not reflected in the subpopulations of CD4 or CD8 T lymphocytes, since the naive and memory CD4 T cells have a similar frequency in terms of receptor expression, with the transition population being at a lower frequency, while the T CD8 has a smaller memory population expressing these receptors, mainly ACKR2 and ACKR3.

### 3.4. Expression of ACKR2, ACKR3 and ACKR4 in T Lymphocytes after Treatment with IL-32, CXCL8 and/or IL-15

CD3 T lymphocytes previously selected and cultured in the presence of IL-32, CXCL18, IL-15, CXCL18+IL-15 or activated with antiCD3:CD28 for 18 h were evaluated for surface or intracellular expression of ACKR2, ACKR3 and ACKR4. The ACKR2 expression was evaluated by confocal microscopy in T cells and its expression was confirmed as demonstrated by Figure 4A. Contrary to what we expected, since we have observed enhancing expression of ACKR2 and ACKR3 at mRNA level after these treatments, the stimulated cells did not differ from the control group in terms of surface expression of ACKR2 (Figure 4B–D), ACKR3 (Figure 4E–G) and ACKR4 (Figure 4H–J) receptors. Surprisingly, after intracellular staining for these same receptors, the frequency of cells positive for all atypical receptors increased up to 100% (Figure 4B,E,H) regardless of the stimulus.

The mean fluorescence intensity (MFI) did not change the expression of ACKRs on the cell surface as demonstrated in the Appendix A. The mean fluorescence intensity (MFI) of the ACKRs also remained similar after intracellular relabeling, even despite a significantly higher level compared to the surface, demonstrated in the Appendix A.

### 3.5. ACKRs mRNA Expression on T Cells Was Described on Databases

Reinforcing our findings, and according to Schmiedel [27] and Monaco [28] datasets, on The Human Protein Atlas (https://www.proteinatlas.org) website accessed on 2 November 2022, among immune cells, different subsets of T lymphocytes present normalized gene expression values (nTPM) for ACKR2, ACKR3 and ACKR4 as demonstrated by Figure 5. In addition, at protein levels, all the 03 ACKRs receptors were described at Human Protein Atlas (https://www.proteinatlas.org—section for subcellular compartment) to be expressed in cell membrane and in cytoplasmic vesicles in few human lineage cells, corroborating our findings for the high values found in the intracellular staining.

### 3.6. Post-Sorting Migration Assay

CD3 T lymphocytes obtained from healthy donors were sorted for three distinct subpopulations ACKR3, CXCR4 or double positive for ACKR3+CXCR4 and placed separately in a transwell insert for the migration assay (Figure 6A). After a 24 h period, the lymphocytes were quantified. CD3/ACKR3 T lymphocytes migrated more towards the CXCL12 chemokine compared with CD3/CXCR4 T lymphocytes and CD3 T lymphocytes that express both ACKR3 and CXCR4 (Figure 6B).

## 4. Discussion

Our data demonstrated that, in general, peripheral blood leukocytes from healthy donors express ACKR2, ACKR3 or ACKR4. Lymphoid cells (T, B and NK cells) expressed less ACKRs than the myeloid ones (granulocytes and monocytes), therefore suggesting an intermediary level of expression among leucocytes.

By specifically analyzing the T lymphocyte populations, we found that CD3 lymphocytes express more ACKR2 or ACKR3 than ACKR4. This phenotypic characteristic is probably related to the modulation of pro-inflammatory chemokines that are ligands of this receptor [12].

ACKR2 was described to be expressed in dermal epithelial cells, lung, intestine, placenta [8,29] T lymphocytes [13,14,30], endothelial lymph cells, B lymphocytes [6,8,29], dendritic cells [6], syncytiotrophoblasts, trophoblasts [8,15,31] and some subsets of macrophage [31]. Even though the ACKR2 expression has been demonstrated in a variety of cells, the cells used were mainly from mice or human lineage cells, and ACKR2 expression was not previously described in human T cells. Our results are reinforced by the mRNA expression in human T cells found in databases.

ACKR3 is described as expressed on hematopoietic cells, neurons, mesenchymal cells, endothelial cells, and cancer cells [31,32,33], CD4 T lymphocytes, NK cells, dendritic cells, neutrophils [31], monocytes, and macrophages [33,34]. The expression of ACKR3 in CD4 and CD8 T lymphocytes was previously demonstrated by Hartmann et al., but not in the data presented by Balabanian et al. Our data corroborate the findings of Hartmann et al. and are supported by mRNA expression found in these in databases. However, our outcomes disagree with data provided in the study by Balabanian et al. [17,35].

Balabanian et al. used a primary antibody anti-ACKR3 clone:9C4 and a secondary horse anti-mouse antibody fluorescent (Texas red) to search for ACKR3 in T cells. In their study, the expression was evaluated by fluorescence microscopy [17]. Hartmann et al. used a primary antibody anti-ACKR3 clone 358426 and a secondary anti-mouse fluorescent (Alexa 488), the analysis was performed by both fluorescence microscopy and flow cytometry [35]. On the other hand, our study used a fluorescent primary antibody (BV421) clone 10D1, which makes it difficult to compare with the other two studies.

ACKR4 has already been described as expressed in stromal cells, endothelial epithelial cells [24], thymic epithelial cells [5,6,24], bronchial cells, keratinocytes, and T lymphocytes [5]. The expression of ACKR4 in CD4 and CD8 T lymphocytes was lower compared with ACKR2 and ACKR3. Our data agree with Heinzel et al. findings, who also described ACKR4 expression in both CD4 and CD8 T lymphocytes, even though their samples were derived from mice, while our samples were derived from human blood [36].

The evaluation of ACKR2 expression in T lymphocytes in CD4/CD8 T and their subsets of naive, transition and memory indicate that the CD8 naive subset expresses more ACKR2 compared to the CD8 memory subset. Yet we noticed that ACKR2 was more expressed in the CD4 transition and memory subsets than in naive. However, the expression of ACKR2 in CD8 was higher in transition than in memory subsets [37,38].

Additionally, when evaluating the expression of ACKR3 on T cells subsets, our data indicates ACKR3 higher expression on CD4 naive and memory subsets than in the transition ones. Accordingly, CD8 naive and transition subsets also displayed higher expression of ACKR3 compared with memory ones. As for ACKR4 expression, there was higher expression in memory and naive CD4 subsets compared with transitional ones [39].

For memory subsets of T lymphocytes, we observed that CD4 expressed higher amounts of ACKR2, ACKR3 and ACKR4 compared to CD8. This characteristic was also observed in naive subset, for ACKR2 and ACKR3 expression, and no differences for ACKR4 between CD4 and CD8 was seen [40].

According to Caccamo et al., when activated, naive T lymphocytes tend to lose their properties and start to have effector and memory characteristics. This process is characterized by the transition subset, which was also evaluated in the present study [39,41]. During ACKR2, ACKR3 and ACKR4 evaluation we observed that CD4 lymphocytes expressed more of these receptors in naive and memory subsets compared with transition ones. This difference may have occurred due to some transcriptional alteration that induced the internalization of ACKRs during the phenotype transition. Here, we hypothesize that the stimulus generated in lymphocytes after their contact with antigens may have contributed to the internalization of the receptor and lead to a phenotype variation during lymphocyte activation/transition [42,43].

For CD8 T cells, we observed variations only for ACKR2 and ACKR3 receptors on naive and transition compared to memory subset. This observation indicates that after activation, memory subset partially loses ACKR2 and ACKR3 expression compared with naive and transition subsets. This situation may be due to changes in the phenotypic characteristics of memory subset [44].

We evaluated whether the cultivation of CD3 T lymphocytes stimulated with IL-32, CXCL8 and IL-15 or CXCL8+IL-15 could favor the protein expression of ACKRs. The expression of ACKR2, ACKR3 and ACKR4 in CD4 and CD8 T lymphocytes treated with these stimuli was not modulated compared with control, and the same was observed in the intracellular compartment.

In the intracellular evaluation, the expression of atypical chemokine receptors was highly present in all lymphocytes. The intracellular expression of ACKRs did not change with the treatments used and did not vary between fresh blood or after laboratory manipulation.

Our results agree with those of Sjöberg et al., 2020, who reported the expression of ACKR2 and ACKR3 to be primarily found intracellularly and in continuously cycling from endosomal compartments to the cell surface. In addition, such cyclic expression is a typical characteristic of scavenger receptors [45].

Thus, the atypical receptors seem to be produced and maintained in the cytosol of lymphocytes, suggesting a post-transcriptional control of its expression in cell membrane and probably dependent on stimuli. The treatments used in this study were not able to increase receptor production in the cytosol or increase their expression in the cell membrane, even though we have observed ACKRs mRNA enhanced expression by these treatments. However, much remains to be elucidated regarding the stimulus that would translocate these receptors from the cytosol to the cell membrane.

To functionally access ACKR3, we have investigated the action of chemokine CXCL12 on CXCR7 (ACKR3), CXCR4 (typical receptor), and on T cells that express both CXCR7 and CXCR4. These three populations were separated by sorting and evaluated by migration assay to CXCL12 gradients. T cells that expressed only CXCR7 (ACKR3) were the lymphocytes that most migrated. Although CXCR7 (ACKR3) is known as a chemokine scavenger receptor, several studies claim that its signaling only occurs via β-arrestin [46] for the subsequent degradation of the chemokine ligand [21,47]. Nevertheless, our functional assay indicates that a signaling provided by G protein coupled to those receptors may have occurred to maintain the migratory profile of these T cells. This result agrees with the Melo et al. study that demonstrated a role for this receptor in hematopoietic stem cells migration, and the way in which their silencing in U937 cells with lentivirus-mediated shRNA impacted these cells’ migration ability [20].

The expression of atypical receptors has been associated with immune responses as innate and adaptive. ACKRs seems to be recycled by pro-inflammatory signaling in the milieu and to control the inflammation by their scavenger role [45]. In addition, inflammatory CC chemokines have long been associated with cancer. ACKR2 expressed on lymphatics vessels was reported by Vetrano et al. to control intestinal inflammation and the inflammation associated with the development of colon cancer [48]. In addition, several small molecules are known to interact with ACKR3. CCX771 is one of the ACKR3 modulators described to induce β-arrestin recruitment to the receptor and it was reported to inhibit tumor growth, lung metastasis, and tumor angiogenesis in in vivo tests [49,50]. Our data have demonstrated the ACKRs expression in different populations of immune cells, placing these cells not only as immune response effectors, but also as players in inflammatory regulation, a fact that deserves further investigation.

## 5. Conclusions

In this paper, it was demonstrated that circulating hematopoietic cells express ACKR2, ACKR3 and ACKR4. These receptors are more expressed by myeloid cells (monocytes and granulocytes) than by lymphoid cells (NK, T and B cells). T cell subsets differentially express these receptors, and they are highly expressed in the intracellular compartment, suggesting a post-transcriptional regulation, highlighting their importance. For ACKR3/CXCR7, the functional assays showed a signaling beyond the classical β-arrestin pathway.

## Figures and Tables

**Figure 1 cells-11-04099-f001:**
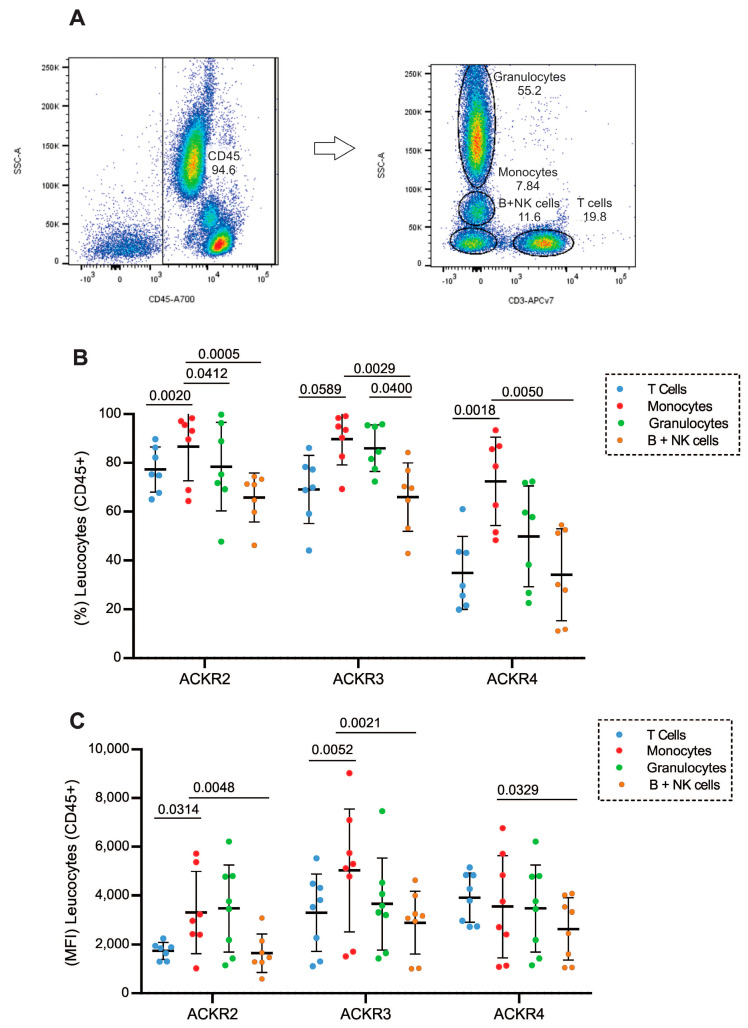
Expression of atypical receptors (ACKR2, ACKR3 and ACKR4) on circulating leukocytes (**A**) Flow cytometry strategy for selection of leukocyte populations based on their CD45 expression, T cells for their CD3 expression, and based on granularity and CD3 exclusion the selection of granulocytes, monocytes and B cells/NK cells; (**B**) expression of ACKR2, ACKR3 and ACKR4 in T lymphocytes (blue circles), monocytes (red circles), granulocytes (green circles) and B cells/NK cells (orange circles) from peripheral blood from healthy donors; (**C**) Mean Intensity of Fluorescence (MFI) of ACKR2, ACKR3 and ACKR4 in T lymphocytes (blue circles), monocytes (red circles), granulocytes (green circles), and B cells/NK cells (orange circles) from peripheral blood from healthy donors (n = 7).

**Figure 2 cells-11-04099-f002:**
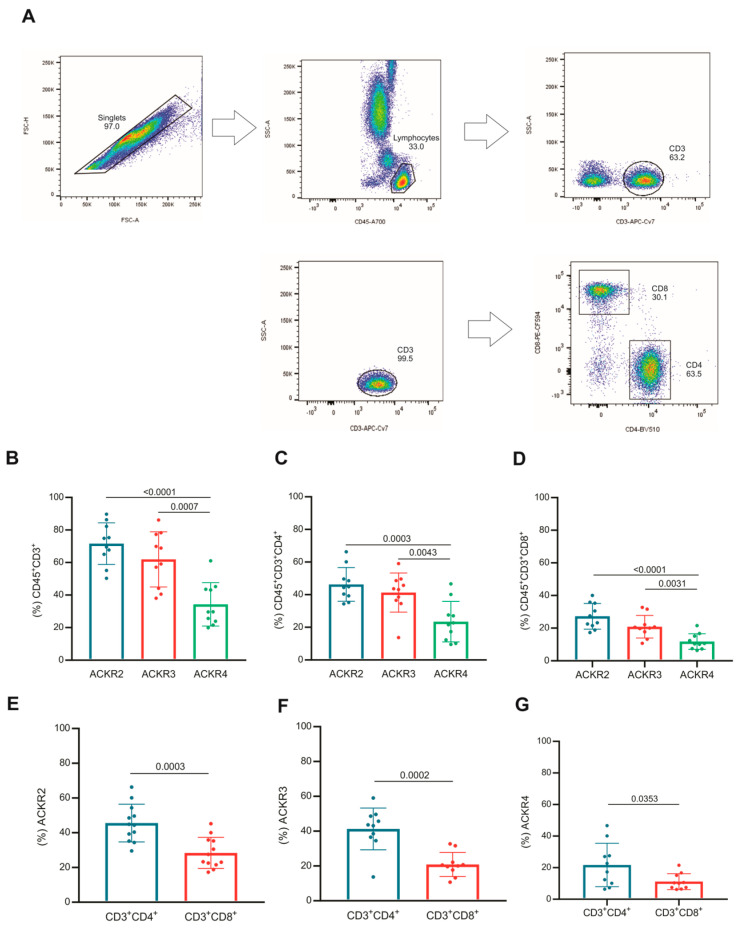
T lymphocytes and subsets that express atypical chemokine receptors ACKR2, ACKR3 and ACKR4. (**A**) Flow cytometry strategy for selection of T lymphocytes was based on their CD45 and CD3 expression, and the T lymphocytes subsets were defined by the expression of CD4 or CD8. (**B**) Expression of ACKR2 (blue), ACKR3 (red) and ACKR4 (green) in populations of CD3 T lymphocytes, (**C**) CD4 T and (**D**) CD8. Expression of ACKR2 (**E**), ACKR3 (**F**) and ACKR4 (**G**) on CD4 (blue) and CD8 (red) T lymphocytes (n = 10).

**Figure 3 cells-11-04099-f003:**
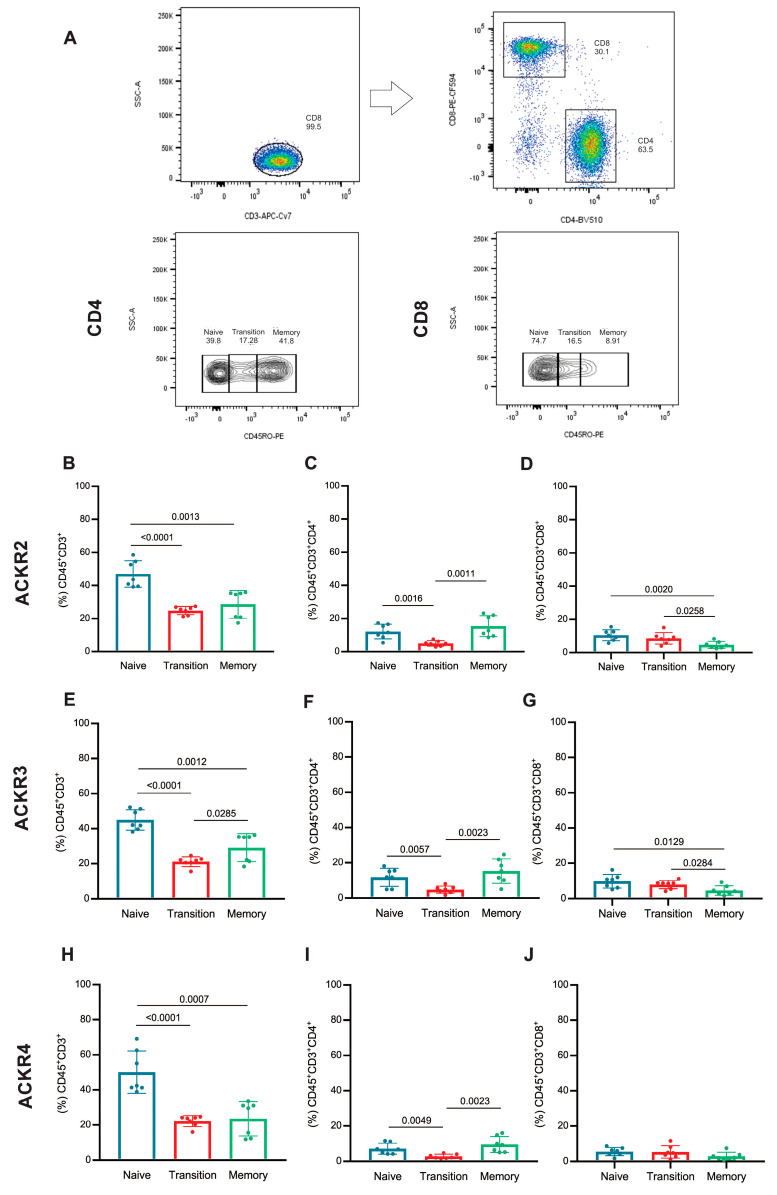
Frequency of naive, transition and memory T lymphocytes expressing ACKRs: (**A**) flow cytometry strategy for selection of T lymphocytes was based on their CD45 and CD3 expression, and the T lymphocytes subsets were defined by the expression of CD4 or CD8, and by expression of CD45RO these lymphocytes subsets were classified as memory (CD45RO+), naive (CD45RO−) and transition (CD45ROint). ACKR2 (**B**) frequency in CD3 T lymphocytes, (**C**) frequency in CD3:CD4 T lymphocytes and (**D**) frequency in CD3:CD8 T lymphocyte subsets, ACKR3 (**E**) frequency in CD3 T lymphocytes, (**F**) frequency in CD3:CD4 T lymphocytes and (**G**) frequency in CD3:CD8 T lymphocyte subsets and ACKR4 (**H**) frequency in CD3 T lymphocytes, (**I**) frequency in CD3:CD4 T lymphocytes and (**J**) frequency in CD3:CD8 T lymphocyte subsets. T lymphocytes naive (blue), transition (red) and memory (green) (n = 7).

**Figure 4 cells-11-04099-f004:**
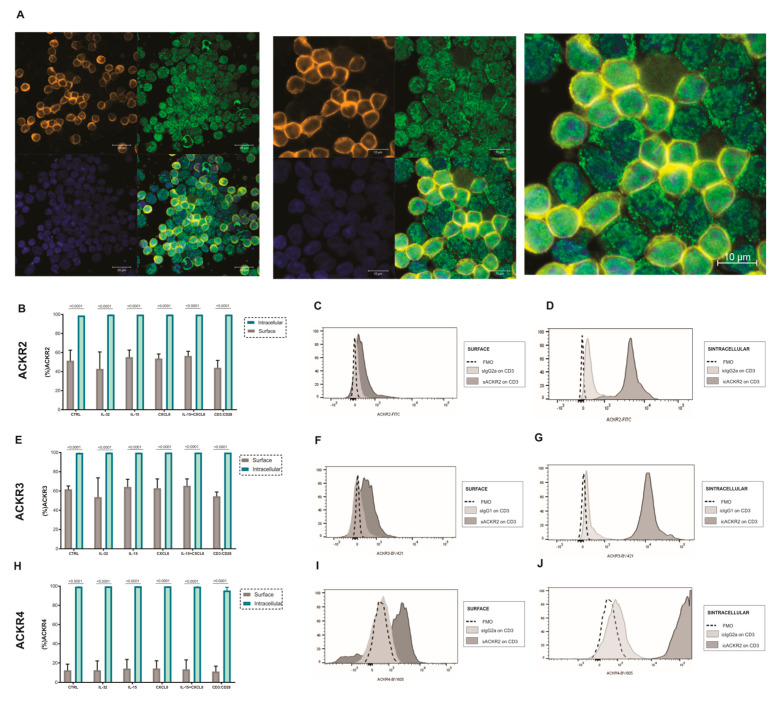
Surface and intracellular expression of ACKRs after different stimuli. Atypical chemokine receptor expression assessed by flow cytometry and confocal microscopy in lymphocytes obtained from peripheral blood of healthy donors. ACKR2 (**A**) confocal microscopy of T cell stained with ACKR2-FITC (green), CD3-PE-CF594 (orange) and DAPI (blue) for nuclei (40× and 62×). (**B**) Graph comparing ACKR2 expression after different stimuli on the lymphocytes surface and in the intracellular compartment, (**C**,**D**) Representative data showing an overlaid histogram of FMO, IgG2a and ACKR2 surface and intracellular expression. ACKR3 (**E**) graph comparing ACKR3 expression after different stimuli on the lymphocytes surface and in the intracellular compartment. (**F**,**G**) Representative data showing an overlaid histogram of FMO, IgG1 and ACKR3 surface and intracellular expression. ACKR4 (**H**) graph comparing ACKR4 expression after different stimuli on the T lymphocytes surface and in the intracellular compartment. (**I**,**J**) Representative data showing an overlaid histogram of FMO, IgG2a and ACKR4 surface and intracellular expression (n = 4).

**Figure 5 cells-11-04099-f005:**
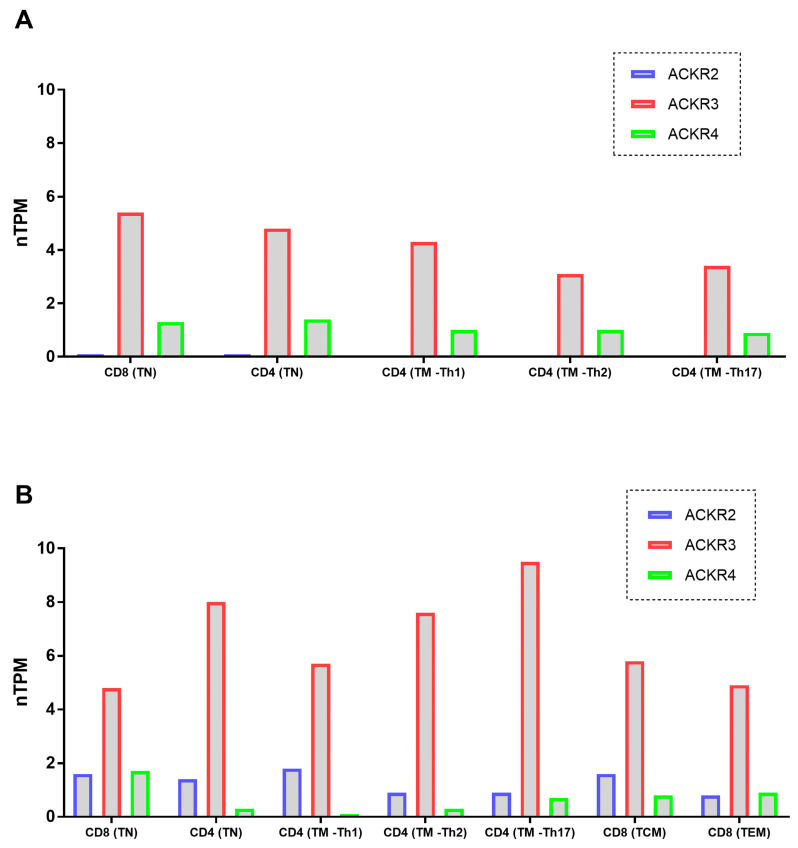
ACKR2, ACKR3 and ACKR4 mRNA expression on T lymphocytes populations according to public databases (**A**) Schmiedel datasets and (**B**) Monaco datasets. Legend: TN—Naive T cell; TM—Memory T cell; TCM—Central Memory T cell; TEM—Effector Memory T cell.

**Figure 6 cells-11-04099-f006:**
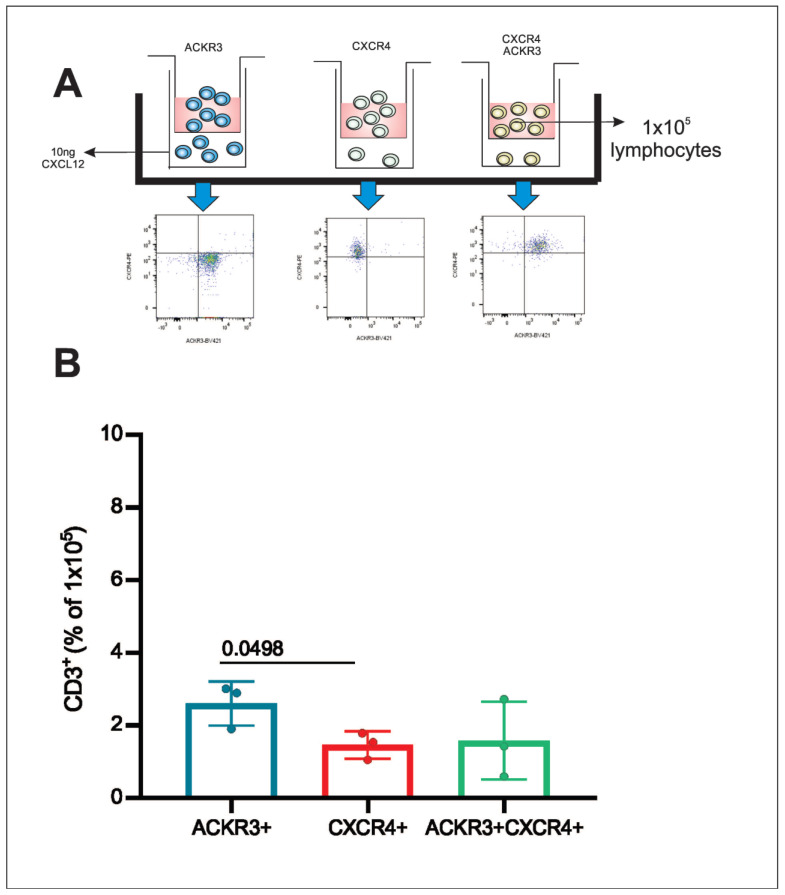
Migration of CD3 T lymphocytes after 24 h stimulation with CXCL12. (**A**) CD3 T lymphocytes from peripheral blood from healthy donors (n = 3) were isolated by flow cytometer cell sorting according to the surface expression of ACKR3, CXCR4 or ACKR3+CXCR4, and cultured in the presence of CXCL12 (under chamber) for 24 h in a transwell culture plate. (**B**) Afterwards, the cells that migrated were collected and the absolute number obtained by flow cytometry was also analyzed for the expression of ACKR3 (blue), CXCR4 (red) or ACKR3+CXCR4 (green). The percentage of CD3 T lymphocytes expressing ACKR3 or CXCR4 or ACKR3+CXCR4 that migrated via transwell towards the CXCL12 ligand was calculated in relation to the initial number of lymphocytes (1 × 10^5^) placed on top of the insert (n = 3).

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
