# Peer review of "Evaluation of Atypical Chemokine Receptor Expression in T Cell Subsets"

_cells, 2022, doi:10.3390/cells11244099_

Round 1

Reviewer 1 Report (Previous Reviewer 2)

I must still take issue with the data in Supplementary Figure 25. The authors have simply removed an inconvenient piece of data without justification. This is not acceptable research practice and must be added back before publication.

In addition, although no data values are equal to, or below, zero, nonetheless that mean and SD of the data may still overlap with zero. The one sample t test will show any significance of the difference with zero. If this test shows no significance, then the authors cannot claim receptor expression.

Author Response

Reviewer 1

I must still take issue with the data in Supplementary Figure 25. The authors have simply removed an inconvenient piece of data without justification. This is not acceptable research practice and must be added back before publication. In addition, although no data values are equal to, or below, zero, nonetheless that mean, and SD of the data may still overlap with zero. The one sample t test will show any significance of the difference with zero. If this test shows no significance, then the authors cannot claim receptor expression.

Answer: I would like to apologize; I have not understood your request at the first time I have answered it and I was not clear about the data removal, I hope now to be able to clarify your doubts.  Our intention was not to be arbitrary; it was only to demonstrate that there was an outlier sample that made the standard deviation very high for MFI of ACKR3. The outlier sample was mathematically determined, since the value of this sample (outlier) is 3 times above of the median value for all the samples. For example, for  the control sample the median was 26,264, while this sample was 177,220 which is more than 3 times the median value of all samples. In addition, in the program GraphPad Prisma, the analysis for outlier identification using Grubbs’ for an alpha = 0.5 identified this same sample as an outlier and removed this data from the analysis. This was the reason that we have previously removed data from the analysis.

However, we have returned this value (outlier) to the graph included in the supplementary material in Fig. 2S and since there is an outlier, we have performed a statistical nonparametric test as Mann-Whitney and the result was significant when compared to zero (p=0.0286), as the analysis demonstrated below.

Table Analyzed

MFI_IC_ACKR3

Column F

Zero

vs.

vs,

Column A

CTRL

Mann Whitney test

P value

0,0286

Exact or approximate P value?

Exact

P value summary

*

Significantly different (P < 0.05)?

Yes

One- or two-tailed P value?

Two-tailed

Sum of ranks in column A,F

26 , 10

Mann-Whitney U

0

Reviewer 2 Report (Previous Reviewer 1)

I am satisfied with the revisions made.

Can the Authors please do a final, careful proof-read and also make sure they check all the Figure numbering and referencing is correct as I detected several errors. For example the new figure 4A is not referenced, and the reference 'Figs 4A, 4C, 4E' should read 'Figs 4B, E, H'.

Author Response

Can the Authors please do a final, careful proof-read and also make sure they check all the Figure numbering and referencing is correct as I detected several errors. For example, the new figure 4A is not referenced, and the reference 'Figs 4A, 4C, 4E' should read 'Figs 4B, E, H'.

Answer: Thank you for pointing out this problem. We have revised the entire manuscript and we hope to have corrected all these points.

Round 2

Reviewer 1 Report (Previous Reviewer 2)

The authors have made a satisfactory response to my point.

This manuscript is a resubmission of an earlier submission. The following is a list of the peer review reports and author responses from that submission.

Round 1

Reviewer 1 Report

This is an interesting survey of atypical chemokine receptors on T cells. However the methodology requires improvement and the paper also requires further English language editing before it can be published.

Specific points:

The introduction is too brief in describing prior work done with these ACKR and T cells in particular. The authors should explicitly state what has been done before in the context of T cells and whether this was in mouse or human systems.  Otherwise the reader is left wondering what is novel about the work.

For the Methodology and Results:

The methods state donors n=40, but in each plot only 7-8 donors are shown.  This needs to be explained.

The authors should consult publicly available RNA and protein datasets to give independent support to their finding of ACKR expression on T cells.

Representative histograms or dotplots for all ACKR flow cytometric analysis must be shown to allow the reader to assess the quality of the staining.

A secondary method of analysis such as immuno-fluorescence on cytospins and ideally tissue sections such as tonsil tissue should ideally be performed to better assess the surface vs. cytoplasmic staining.

Why were CXCL8, IL-15 and IL-32 chosen to stimulate the cultures, this highly unusual approach should be justified explicitly in the text. Direct antiCD3/CD28 stimulation must be performed alongside to validate this method.

An FMO control is shown for the intracellular staining but the correct control would be an fluorophore-conjugated isotype control. IC staining is known for its high background, and the high level of ACKR staining could simply be due to antibody trapped within the cell in a non-specific manner during permeabilization. Intracellular staining should be further validated by IF as noted above.

Functional analysis should be extended to all ACKRs, not just ACKR3. Otherwise we can’t determine i the relatively low level expression of ACKR on T cells has any biological significance at all.

In the discussion the authors need to better discuss the biological significance of the expression of these receptors on T cells. What does it mean for their function in healthy or disease states? Is it a target for therapeutic intervention?

Author Response

This is an interesting survey of atypical chemokine receptors on T cells. However, the methodology requires improvement and the paper also requires further English language editing before it can be published.

Answer: Thank you, for the careful review of the article. We have improved the English and reviewed the introduction and methodology of the article. We also have included new analysis and controls.

Specific points:

The introduction is too brief in describing prior work done with these ACKR and T cells in particular. The authors should explicitly state what has been done before in the context of T cells and whether this was in mouse or human systems.  Otherwise the reader is left wondering what is novel about the work.

Answer: We have included in the introduction a report about the expression of these receptors in human cells, as follows:

“Regarding the protein expression of ACKRs in the membrane of human cells, the ACKR2 expression was only demonstrated in human lineage cells, but its expression was not described previously in human T cells. The ACKR3 was found expressed on human T cells (17), to date, there is no report about its expression on human T cells subsets. In addition, the ACKR4 was only described in mouse T cells and human lineage cells, this receptor was not previously described to be expressed in human T cells.”

For the Methodology and Results:

The methods state donors n=40, but in each plot only 7-8 donors are shown.  This needs to be explained.

Answer: We have included in the methodology an explanation about the samples as follows:

“These 40 samples were distributed among different experiments, an initial 08 samples were used in order to establish the antibodies titration, compensation and to set up cell sorting experiments. Next, 07 samples were freshly stained for the ACKRs, 10 samples were stained for ACKRs oor T cells subsets CD4 and CD8, 07 samples were used for ACKRs on T cells subsets (memory/naïve), 04 samples were used for surface and intracellular expression of ACKRs, 04 samples were used for test different stimuli and another 04 for cell sorting and chemotaxis.”

The authors should consult publicly available RNA and protein datasets to give independent support to their finding of ACKR expression on T cells.

Answer: We have consulted databases  and the findings were included in the article methodology (item 2.9) and results (item 3.5)

Representative histograms or dot plots for all ACKR flow cytometric analysis must be shown to allow the reader to assess the quality of the staining.

Answer: We have representative dot plots for all ACKRs for all populations analyzed (Monocytes, Granulocytes, B/NK cells and T cells) as a Fig S1 in supplemental material

A secondary method of analysis such as immuno-fluorescence on cytospins and ideally tissue sections such as tonsil tissue should ideally be performed to better assess the surface vs. cytoplasmic staining.

Answer: We performed a confocal photo for the ACKR2-FITC intracellular staining vs staining with IgG2a intracellular, to demonstrating that the staining was not unspecific, and micrography was included in Fig 4.

Why were CXCL8, IL-15 and IL-32 chosen to stimulate the cultures, this highly unusual approach should be justified explicitly in the text. Direct antiCD3/CD28 stimulation must be performed alongside to validate this method.

Answer: We have chosen CXCL8, IL-15 and IL-32 for stimulate the cultures, because we have an unpublished data that evidence the enhanced mRNA expression of ACKRs on T cells after those treatments. So, we intended to correlate with our previous data, however, we observed that the mRNA is not related with the protein levels on the surface of cells for these receptors.

An FMO control is shown for the intracellular staining but the correct control would be an fluorophore-conjugated isotype control. IC staining is known for its high background, and the high level of ACKR staining could simply be due to antibody trapped within the cell in a non-specific manner during permeabilization. Intracellular staining should be further validated by IF as noted above.

Answer: We have added the isotype control to the ACKRs for surface and intracellular staining, and we do observe a difference in background, but this difference is not impacting our previous  results. We have added a representative figure overlapping the FMO/IgG and the ACKRs staining in the Fig. 4.

Functional analysis should be extended to all ACKRs, not just ACKR3. Otherwise we can’t determine i the relatively low level expression of ACKR on T cells has any biological significance at all.

Answer: We have selected ACKR3 to perform the functional assays because this is the less promiscuous receptor between all the 03 receptors. The ligands for ACKR3 are only CXCL12 and CXCL11, and there is only CXCR4 as conventional receptor for competing for the same chemokine. Using this receptor, we were able to use the cell sorting  to better to decipher the results. 

The other 02 receptors have to many ligands. ACKR2 has as ligands CCL2, CCL3, CCL4 CCL5, CCL7,  CCL8, CCL11, CCL12, CCL13, CCL17 and CCL22, and several conventional receptors to compete with as CCR1, CCR2, CCR3, CCR4 and CCR5. The ACKR4 also has many ligands as CCL19, CCL21, CCL25 and CXCL13 and several conventional receptors to compete as CCR6, CCR7, CCR9 and CXCR5. making difficulty to interpret data generated by a functional study.

In the discussion the authors need to better discuss the biological significance of the expression of these receptors on T cells. What does it mean for their function in healthy or disease states? Is it a target for therapeutic intervention?

Answer: We have included information for their role in inflammation in the discussion final paragraph as follows:

The expression of atypical receptors has been associated with immune responses as innate and adaptive. ACKRs seems to be recycled by pro-inflammatory signaling in the milieu and to control the inflammation by their scavenger role (45). In addition, inflammatory CC chemokines have long been associated with cancer. ACKR2 expressed on lymphatics vessels was reported by Vetrano et al to control intestinal inflammation and the inflammation-associated with the development of colon cancer (50). In addition, several small molecules are known to interact with ACKR3. CCX771 is one of the ACKR3 modulators, described to inducing β-arrestin recruitment to the receptor and was reported to inhibit tumor growth, lung metastasis and tumor angiogenesis in in vivo tests (51, 52). Our data have demonstrated the ACKRs expression in different populations of immune cells, placing these cells not only as immune response effectors, but also as players in inflammatory regulation, which deserves to be further investigated.

Reviewer 2 Report

This is a simple study of receptor expression in human T cells, with a small functional element. It makes a minor contribution to our knowledge in this area.

I am concerned, however, that there may be major issues with the methodology.

Firstly, no isotype control antibodies appear to have been used, meaning that no account is taken of non-specific staining. This is always important but particularly so when intracellular staining is measured. The non-specific staining is much higher when cells are permeabilised, and is highly cell-type dependent. Before publication, the authors must show that isotype control antibody binding does not vary. This is a separate issue from the 'FMO' technique used to assess interference between channels in flow cytometry.

Secondly, most expression data is shown as percentage positive cells, but with no indication of the actual expression levels (e.g. MFI). MFI data should be shown alongside percentage expression so that an asessment can be made of the amount of receptor on each cell type.

Finally, where expression levels are shown (Fig 4), these are apparently normalised to intracellular levels. We cannot see what the original expression levels of the receptors are. This data should be shown as separate values for intra- and extracellular expression, after adjustment for isotype control staining.

Minor issues:

The English language needs correcting, as sense is sometimes lost.

Author Response

This is a simple study of receptor expression in human T cells, with a small functional element. It makes a minor contribution to our knowledge in this area.

I am concerned, however, that there may be major issues with the methodology.

Firstly, no isotype control antibodies appear to have been used, meaning that no account is taken of non-specific staining. This is always important but particularly so when intracellular staining is measured. The non-specific staining is much higher when cells are permeabilised, and is highly cell-type dependent. Before publication, the authors must show that isotype control antibody binding does not vary. This is a separate issue from the 'FMO' technique used to assess interference between channels in flow cytometry.

Answer: We have added the isotype control to the ACKRs for surface and intracellular staining, and we do observe a difference in background, but this difference is not impacting our previous  results. We have added a representative figure overlapping the FMO/IgGs and the ACKRs staining in the Fig. 4.

Secondly, most expression data is shown as percentage positive cells, but with no indication of the actual expression levels (e.g. MFI). MFI data should be shown alongside percentage expression so that an asessment can be made of the amount of receptor on each cell type.

Answer: We have included the MFI graph for the expressions in the main figure, and as supplemental material for surface and intracellular staining.

Finally, where expression levels are shown (Fig 4), these are apparently normalised to intracellular levels. We cannot see what the original expression levels of the receptors are. This data should be shown as separate values for intra- and extracellular expression, after adjustment for isotype control staining.

Answer: We have separated the graphs with the percentage levels and MFI levels and added this information in the supplemental material as figure S2.

Minor issues:

The English language needs correcting, as sense is sometimes lost. Answer:

Answer: We have revised the English and corrected the sentences in this manuscript

Round 2

Reviewer 1 Report

The authors have provided acceptable responses to some, but not all, of my original comments.

While some English-language editing has occurred there are still numerous issues, including typos in the revised text (e.g. lines 73, 204, 322, 385 ). Line 82 should read 'human T cells'. Lines 84-85 appear to be directly contradictory. I recommend an external English editor be used.

A database search was performed but the results were not presented in any objective way e.g. via graphical representation.

An alternative method for staining the atypical chemokine receptors was requested however the authors have performed an identical staining method and simply taking a very low-powered microscope image from a tissue culture plate. I do not believe that this was done with a confocal microscope but more likely an epi-fluorescent microscope. The purpose of this requested experiment was to better visualise the location (membrane or intracellular) of the ACKR and support the claim in the manuscript for high-level intracellular expression. This experiment should be performed on cytopspins of PBMC, or similar, to a resolution sufficient to determine the morphology of the cells and spatial location of the ACKR (40x or higher). Ideally a CD3 and DAPI co-stain would be performed. 

An appropriate positive control was requested for the T cell stimulation experiment given the use of an unpublished methodology. This was not performed.

Author Response

Reviewer 1

The authors have provided acceptable responses to some, but not all, of my original comments.

Answer: We hope now to have properly provide all the experiments requested by this reviewer.

Question: While some English-language editing has occurred there are still numerous issues, including typos in the revised text (e.g. lines 73, 204, 322, 385 ). Line 82 should read 'human T cells'. Lines 84-85 appear to be directly contradictory. I recommend an external English editor be used.

Answer: We have revised point by point of the manuscript and sent the manuscript to an external revisor.

Question: A database search was performed but the results were not presented in any objective way e.g. via graphical representation.

Answer: We have included a new figure (Fig. 5) in line 348, demonstrating the findings of the 02 public databases consulted.

Question: An alternative method for staining the atypical chemokine receptors was requested however the authors have performed an identical staining method and simply taking a very low-powered microscope image from a tissue culture plate. I do not believe that this was done with a confocal microscope but more likely an epi-fluorescent microscope. The purpose of this requested experiment was to better visualise the location (membrane or intracellular) of the ACKR and support the claim in the manuscript for high-level intracellular expression. This experiment should be performed on cytopspins of PBMC, or similar, to a resolution sufficient to determine the morphology of the cells and spatial location of the ACKR (40x or higher). Ideally a CD3 and DAPI co-stain would be performed. 

Answer: We have provided a new figure inserted in Fig 4 with staining of PBMC after cytospin, using CD3 PE-CF594 for T cells and DAPI for nuclei and ACKR2-FITC (40x), and one photo (62x). We were not able to perform the confocal microscopy for ACKR3 and ACKR4 (BV421 and BV605), due to the antibodies that are not proper for immunofluorescence.

Question: An appropriate positive control was requested for the T cell stimulation experiment given the use of an unpublished methodology. This was not performed.

Answer: We have provided a control with T cells (CD3) stimulated with antiCD3:CD28, the results were included in the graph with other stimuli IL-32, CXCL18, IL-15, CXCL18+IL-15.

Reviewer 2 Report

Although the authors have now included isotype control antibodies, it is unclear as to how the data from the isotypes has been handled. The Methods and Materials section needs to include the calculations used to derive the MFI values.

Some of the MFI values appear to overlap with 0 and so an additional statistical check should be performed, to show that there is a significant difference from 0. I would suggest a 1 sample t test but other methods may also be appropriate. Only data that shows a significant difference from 0 should be included.

The English has been improved to a considerable extent but some of the new sections may still need editing.

Author Response

Reviewer 2

Question: Although the authors have now included isotype control antibodies, it is unclear as to how the data from the isotypes has been handled. The Methods and Materials section needs to include the calculations used to derive the MFI values.

Answer: MFI values were obtained from FlowJo Software for each AKCR fluorescence, and the background MFI from isotype control was diminished from the original value, this information was included in method section (item 2.3).

Some of the MFI values appear to overlap with 0 and so an additional statistical check should be performed, to show that there is a significant difference from 0. I would suggest a 1 sample t test but other methods may also be appropriate. Only data that shows a significant difference from 0 should be included.

Answer: There was no value equal or below zero for MFI. However, for ACKR3-MFI as demonstrated in supplemental material there was a large variation on the MFI value,  so even though there were no value equal or below zero, the variation demonstrated by the standard deviation display negative values. This have occurred only on ACKR3, due to one sample with very high MFI-values and not by negative values. Consequently, we have considered this sample as an outlier sample and removed it from the ACKR3-MFI analysis. 

The English has been improved to a considerable extent but some of the new sections may still need editing.

Answer: Thanks, we have revised the English and sent to an external reviewer.
